# Usefulness of Magnetic Mallet in Oral Surgery and Implantology: A Systematic Review

**DOI:** 10.3390/jpm12010108

**Published:** 2022-01-14

**Authors:** Francesco Bennardo, Selene Barone, Camillo Vocaturo, Ludovica Nucci, Alessandro Antonelli, Amerigo Giudice

**Affiliations:** 1School of Dentistry, Department of Health Sciences, Magna Graecia University of Catanzaro, Viale Europa, 88100 Catanzaro, Italy; selene.barone@studenti.unicz.it (S.B.); a.giudice@unicz.it (A.G.); 2Private Practice, Via Nicola Serra 80, 87100 Cosenza, Italy; camillovocaturo@gmail.com; 3Multidisciplinary Department of Medical-Surgical and Dental Specialties, University of Study of Campania, Luigi Vanvitelli, 80138 Naples, Italy; ludovica.nucci@unicampania.it

**Keywords:** dental implants, alveolar ridge augmentation, sinus floor augmentation, tooth extraction

## Abstract

This systematic review aimed to answer the question: “Is the use of magnetic mallet effective in oral and implant surgery procedures in terms of tissue healing, surgery outcome, and complication rate compared to traditional instruments?” A literature search of PubMed, Scopus, and Web of Science databases (articles published until 1 October 2021) was conducted, in accordance with the PRISMA statement, using the keywords “magnetic mallet”, “electric mallet”, “oral surgery”, “implantology”, and “dental implant”. Of 252 articles, 14 were included in the review (3 for teeth extraction, and 11 for implant dentistry). Out of a total of 619 dental extractions (256 patients) performed with the magnetic mallet (MM), no complications were reported. Implants inserted totaled 880 (525 patients): 640 in the MM groups (382), and 240 in control groups (133). The survival rate of implants was 98.9% in the MM groups, and 95.42% in the control groups. Seven patients experienced benign paroxysmal positional vertigo after implant surgery, all in control groups. Results are not sufficient to establish the effectiveness of MM in oral and implant surgery procedures. Randomized controlled trials with a large sample size are needed.

## 1. Introduction

Alveolar bone resorption after tooth extraction occurs for both physiological and iatrogenic reasons [1]. Several factors have been related to bone resorption: a patient’s general health and behavior, pre-operative condition, tooth type and location, and post-operative management [2]. Though no extraction technique is completely atraumatic, it has been reported to influence the extent of alveolar bone resorption: conventional tooth extraction techniques, involving the use of elevators, periotomes, and forceps, operate on the principle of socket expansion, and will, therefore, traumatize the alveolar bone to some extent, which is often associated with higher tissue trauma and increased post-extraction bone loss [3]. The loss of the bundle bone is secondary to periodontal ligament fibers and vessels interruption, clot instability, and soft tissue collapse [4]. A traumatic extraction may also lead to other complications, such as fracture of a bony wall, poor healing due to a compromised blood supply, and soft tissue damage [1].

Alveolar ridge remodeling occurs most rapidly in the first six months after tooth extraction, but bone resorption continues throughout life at a slower rate [5]. Preservation of an adequate volume of soft and hard tissues is necessary to obtain functional and aesthetic stability of prosthetic restorations, whether removable or fixed, including dental implants [6]. Several minimally-invasive tooth extraction techniques using new technologies (vertical extraction, piezoelectric, magneto-dynamic) have been described. However, there is no evidence of lower alveolar bone resorption [7].

Residual alveolar ridges’ features affect the opportunity of prosthetic rehabilitation with dental implants [8]. Successful osseointegration of dental implants depends on multiple factors, including bone quantity and quality, primary implant stability, fixtures’ design, and surface characteristics [9]. A reduced amount of bone requires reconstruction or innovative surgical techniques [10]. In cases of horizontal defects, the ridge can be expanded, or wedge implants placed [11]. In cases of vertical bone defects, sinus floor elevation (in the maxilla) or short implants are available techniques [12,13].

In recent decades, companies have proposed numerous innovative surgical devices for oral surgery and implantology. These devices mainly work through laser, piezoelectric, and magneto-dynamic technologies. Nevertheless, there is a lack of clinical evidence regarding their application for dental surgery in some cases.

A timely evaluation of new surgical techniques is necessary to prevent widespread adoption with insufficient evidence, and promote innovation supported by sufficient evidence [14]. The evaluation of medical innovation is challenging because surgical technique, approach, and instrumentation continue to evolve as novel techniques are used in practice [1].

Magneto-dynamic technology exploits the physical principles of electromagnetism to apply controlled forces on a body while minimizing the time of impact. The control and steadiness of the applied forces make the procedures safe for patients and surgeons. William Bonwill patented the first electrified dental mallet in 1873: it was used for cavities’ gold filling [15].

In the 21st century, the Magnetic Mallet (MM) device (Meta Ergonomica, Milan, Italy) exploits magneto-dynamic technology in dental surgery. The MM is composed of a handpiece energized by a power control device, delivering forces by the timing of application. Different inserts could be attached to the handpiece, which pushes a shock wave on its tip according to the surgical procedures. Four force modes are available: 75,90,130, and 260 daN. The time of impact is 80 μs [16,17].

Several authors describe the application of MM in dental extractions, crestal sinus lift, ridge expansion, implant placement, and implant site preparation (osseodensification). However, it is not clear whether there is scientific evidence for these MM applications.

The main objective of this study was to conduct a systematic review of the literature to determine whether the use of MM is effective in oral and implant surgery procedures, and to consider the possible role of MM in reducing failures and complications.

## 2. Materials and Methods

The authors followed the criteria established in the Preferred Reporting Items for Systematic Reviews and Meta-Analyses (PRISMA) guidelines for this review [18].

### 2.1. PICO Question

“Is the use of magnetic mallet effective in oral and implant surgery procedures in terms of tissue healing, surgery outcome, and complication rate compared to traditional instruments?”

### 2.2. Search Strategy

An electronic literature search was performed using the following databases: Medline (using PubMed), Scopus, and Web of Science. Articles published up to 1 October 2021 were included. The keywords used were “magnetic mallet”, “electric mallet”, “oral surgery”, “implantology”, and “dental implant”, using the Boolean operators “AND”, “OR”.

### 2.3. Inclusion and Exclusion Criteria

The following inclusion criteria were applied: (1) any original publication in the English language, (2) studies conducted on humans, (3) the use of the MM for surgical procedures.

The following exclusion criteria were applied: (1) in vitro studies; (2) experimental animal studies; (3) radiological studies without clinical evaluation; (4) cadaver studies; (5) the use of the MM for implant placement only; (6) literature reviews, letters, editorials, doctoral theses, or abstracts.

The reference list of review articles was analyzed to search for other articles not found in the electronic literature search.

### 2.4. Selection of the Studies

The manuscripts selected included prospective studies, retrospective studies, and observational studies. Two authors (FB, SB) conducted database searches independently, and discrepancies were resolved in a consensus meeting with a third reviewer (AA).

### 2.5. Data Extraction

Data was extracted by two reviewers independently (FB, CV). Disagreement was subject to a new evaluation with a third reviewer (AG). The variables extracted from the studies were the following: study design, type of intervention (dental extraction, implant site preparation, ridge expansion, sinus lift), number of patients and sites treated, follow-up duration, outcomes, complications.

### 2.6. Data Analysis

Data in the included studies was analyzed with descriptive statistics: total number of cases, percentage of outcome variables, etc. Data was subdivided by type of intervention. If none of the included studies were set up as a randomized clinical study, the meta-analysis would not be performed. An independent researcher (LN) performed descriptive statistical analysis with the STATA software program (STATA, Release 14; STATA Corporation, College Station, TX, USA).

## 3. Results

The results of the literature search are presented in the PRISMA flow diagram (Figure 1).

### 3.1. Study Selection

The search strategy yielded records (225 from PubMed, 59 from Scopus, 29 from Web of Science); and 10 additional articles were identified through a hand search. After the removal of duplicates, 252 records remained. After title and abstract screening, 44 articles were identified for full-text retrieval and analysis. Of these, 30 did not meet the inclusion criteria (articles not in English language, literature reviews, no use of magnetic mallet), and 1 article’s full-text was not retrieved. The remaining 14 articles were included in the systematic review; the included papers are listed in Table 1 according to study design, and type of intervention. None of the included studies were set up as a randomized clinical trial; therefore, none were included in the quantitative synthesis.

### 3.2. Magnetic Mallet in Teeth Extraction

Three studies related to the use of the magnetic mallet for teeth extraction were included: one retrospective clinical study on the utility of the magnetic mallet in dental extraction [22], one prospective study on dimensional changes of fresh sockets with reactive soft tissue preservation after tooth extraction performed with the magnetic mallet [27], and one retrospective clinical study on the effect of different timings of implant insertion on the bone remodeling volume around patients’ maxillary single implants after tooth extraction performed with the magnetic mallet [23]. A total of 256 patients were recruited in the studies considered. Dental extractions totaled 619: no complications related to teeth extraction performed with the magnetic mallet were reported in these studies (no fracture or loss of cortical bone plate, no signs of inflamed tissue or exposed bone). Statistical analysis was not performed considering the lack of control groups.

### 3.3. Magnetic Mallet in Implant Surgery

Eleven studies related to the use of the magnetic mallet in implant surgery were included: two prospective clinical studies, two retrospective clinical studies, and one observational study on sinus lift performed with the magnetic mallet [19,28,29,31,32]; four prospective clinical studies on implant site preparation performed with the magnetic mallet [20,21,24,30]; and one prospective clinical study and one retrospective clinical study on ridge expansion performed with the magnetic mallet [25,26].

A total of 525 patients were recruited in the studies considered: 382 underwent implant surgery performed with the magnetic mallet, and 133 underwent implant surgery performed with standard techniques. Implants inserted totaled 880: 640 in the magnetic mallet groups, and 240 in control groups. The survival rate of implants after 9–66 months was 98.9% in the magnetic mallet groups, and 95.42% in the control groups.

No patients treated in the magnetic mallet groups experienced benign paroxysmal positional vertigo (BPPV), whereas seven patients in the control groups showed this complication after sinus lift (2), implant site preparation (4), and ridge expansion (1) performed with osteotomes pushed by the hand mallet.

Statistical analysis was not performed considering the non-comparability of the studies’ data included in this review.

## 4. Discussion

Fourteen articles were included in this systematic review: three related to the use of the magnetic mallet for teeth extraction, and eleven related to the use of MM in implant surgery, for a total of 256 and 525 treated patients, respectively.

Results related to surgical procedures performed with MM highlighted no complications related to teeth extraction, an implant survival rate of 98.9% after 9–66 months of follow-up, and no patients experienced BPPV.

Considering that none of the included studies were set up as a randomized clinical trial, these results are not sufficient to establish a superior effectiveness of the magnetic mallet compared to standard procedures.

### 4.1. Magnetic Mallet in Teeth Extraction

The trauma sustained by the periodontium during tooth extraction changes considerably depending on the technique used, ranging from the extraction of a single-rooted tooth using periotomes, elevators, and forceps, to the reflection of a mucoperiosteal flap and bone removal. Bone trauma is difficult to avoid, as the principle of tooth extraction is socket expansion, and even a successful extraction using elevators or periotomes will traumatize alveolar bone to some extent. Although no extraction technique can be completely atraumatic, several minimally-invasive tooth extraction techniques using new technologies (vertical extraction, piezoelectric, magneto-dynamic) have been described [7,33].

The literature analysis highlighted a few studies concerning the use of MM for dental extractions, but only one retrospective clinical study entirely focused on this topic. Crespi et al. reported the extraction of 427 hopeless teeth in 156 patients using MM. They observed no soft tissue healing complication, fracture, or loss of cortical bone plate. The authors described how the axial movements applied on the blade’s tip detached the root from the surrounding alveolar bone, reducing trauma to adjacent bone, and avoiding damages to gingival tissues [22]. In the other studies examined, authors did not report complications in teeth extractions performed with MM [23,27].

Considering the advantages described, the use of MM could simplify the procedures for obtaining stem cells of dental origin [34,35].

In 2009, another device that exploited the advantages of atraumatic extractions of the periotome with mechanized speed was brought to the market without success under the name of “Powertome” [36]. On the contrary, the MM has been very successful, and using the same patent, a device dedicated to dental extractions, called “Easyroot”, has recently been introduced [37].

In the last two years, following the outbreak of the severe acute respiratory syndrome coronavirus 2 (SARS-CoV-2) pandemic, recommendations have been issued to avoid the risk of contagion, which includes minimizing the production of aerosols during dental procedures [38,39,40]. Recently, Chien et al. presented an aerosolizing reducing technique for the extraction of impacted mandibular third molars using a chisel and mallet. The use of the chisel requires tapping with a mallet on the handle to separate the tooth from surrounding bone. This technique can cause unnecessary discomfort for the patient, and may cause BPPV. Furthermore, this technique may also be uncomfortable for the surgeon [41].

Given the concern of aerosol-generating techniques and exposure to SARS-CoV-2, the MM could be used for surgical teeth extraction as a way of minimizing aerosol exposure (authors’ opinion).

### 4.2. Magnetic Mallet in Implant Surgery

Innovative devices and technologies to reduce morbidity, biological, and surgical times are an intense research topic in implant dentistry. Implant-prosthetic rehabilitations are a routine treatment, but a minimal amount of bone, both vertical and horizontal, is necessary at implant placement. Several techniques were described to increase the amount of bone before or simultaneously with fixture insertion [42,43]. These surgical procedures involve using various tools, such as surgical burs, saws, laser, or piezoelectric devices. In the crestal sinus lift and ridge-splitting procedures, after corticotomy, the osteotome/chisel is pushed by a hand mallet to provoke a greenstick fracture of the bone with a space-making effect that could be filled with biomaterials or with dental implants at the same time. This technique was also used for implant site preparation. However, the tapping of osteotomies/chisels with a hand mallet can induce BPPV in some cases. The MM handpiece imparts a longitudinal movement along the central axis of the osteotome/chisel that acts upon and forces the internal wall of the hole outward radially, resulting in controlled fracture and displacement of the cortical bone, and greater bone tissue density along the walls [25,28,30]. Furthermore, the MM is currently the gold standard for blade- and wedge-implant insertion [11,44].

The literature analysis highlighted several studies concerning the use of MM for implant surgery, but focused on different types of interventions.

In 2012 and 2013, Crespi et al. reported the successful insertion of 130 dental implants after sinus floor elevation performed with MM in 72 patients, with a survival rate of 98% after two years of follow-up. The results of both prospective studies allowed the authors to evaluate MM as a fast and accurate device in crestal sinus lift, with minor patient discomfort compared to the traditional technique with a hand mallet and osteotomes [28,31]. In the other studies examined, the authors confirmed the validity of MM in crestal sinus lift [19,29,32].

In 2014 and 2015, Crespi et al. reported the successful insertion of 152 dental implants after ridge expansion performed with MM in 59 patients, with a survival rate above 98% after two years of follow-up. The authors stated that segmental ridge split with MM represents a safe, predictable, and comfortable surgical procedure, and it does not lead to bone injury or overheating [25,26].

Crespi et al., between 2012 and 2016, conducted four prospective clinical studies on implant site preparation performed with MM, reporting successful insertion of 218 dental implants in 140 patients, with a survival rate not less than 96% after 24–36 months of follow-up [20,21,24,30].

The survival rate of implants placed on patients treated with MM was 98.9% after 9–66 months of follow-up, and 95.42% in the control groups. No patients treated with MM in the included studies experienced BPPV, probably due to the application of controlled forces in very short, defined intervals. In contrast, seven patients in the control groups experienced BPPV after sinus lift (2), ridge expansion (1), and implant site preparation (4) performed with osteotomes pushed by a hand mallet. Besides, there are biases in the articles examined in the review, concerning the type of study, the presence or absence of control groups, the characteristics of the patients included, and the dissimilar protocols used (flap or flapless surgery, delayed or immediate implant insertion).

Feher et al. evaluated resonance frequency analysis (RFA) values of fixtures placed in condensed bone with implant site preparation performed with MM after drilling a pilot hole of 2.2 mm in diameter, and 8 mm in depth, both in edentulous ridges and in fresh sockets. Their main finding is that mean RFA values correlated with peak insertion torque (PIT) in both uncondensed and condensed bone in edentulous jaws. Vice versa, in fresh extraction sockets, bone condensation led to a loss of correlation between RFA and PIT: implant site preparation with MM led to higher mean RFA, but not higher PIT values. The effects of bone condensation are somewhat complex, and need cautious interpretation and further investigations [45].

## 5. Conclusions

The use of MM may be helpful in oral and implant surgery procedures because of its safety, predictability, speed, and comfort of use. Considering limitations (no studies eligible for meta-analysis, no randomized studies, no multicentric studies, small samples, different protocols), the results of this systematic review are not sufficient to prove MM’s effectiveness in dental surgery.

Further randomized controlled studies are needed to establish whether the use of MM could, on the one hand, significantly reduce the complication rate after oral surgery procedures, and, on the other, improve tissue healing and long-time implant survival.

## Figures and Tables

**Figure 1 jpm-12-00108-f001:**
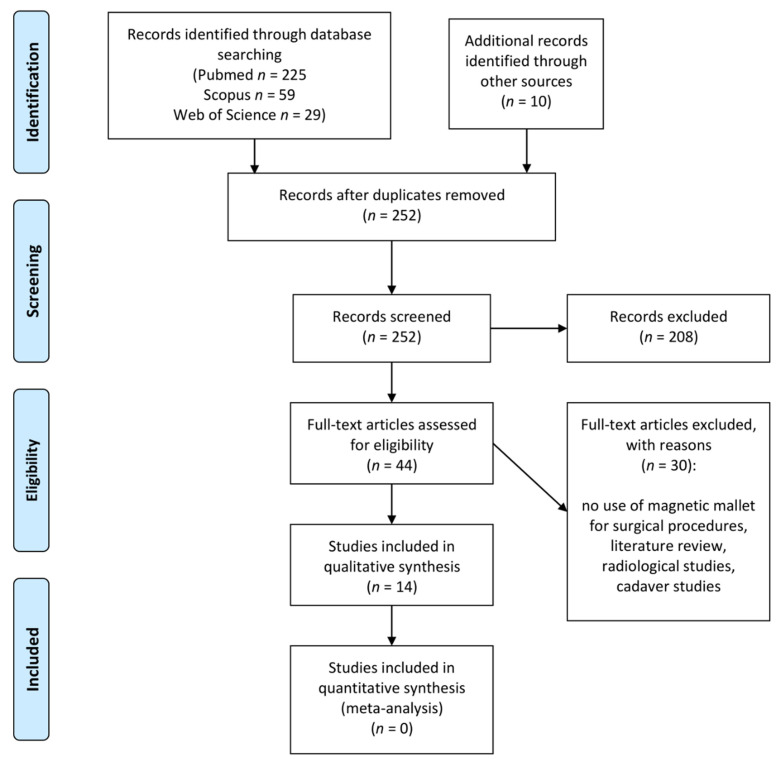
PRISMA flow diagram.

**Table 1 jpm-12-00108-t001:** Specific features and outcomes of the studies included.

Reference	Type of Intervention	Study Design	Number of Patients Treated	Number of Sites (Implants/Teeth) Treated	Follow-Up	Outcomes	Complications
Crespi et al., 2012 A [19]	Sinus lift	Prospective	40 MM 40 CTR	60 MM 60 CTR	6, 12, 24 m	Survival rate of 98.33% (both groups)	2 BPPV in CTR group
Crespi et al., 2012 B [20]	Implant site preparation	Prospective	25 MM 25 CTR	69 MM 69 CTR	6, 12, 24 m	Survival rate of 97.1% (MM) and 92.75% (CTR)	2 BPPV in CTR group
Crespi et al., 2013 A [21]	Sinus lift	Prospective	32	70	2, 4, 12, 24 m	Survival rate of 98.57%	-
Crespi et al., 2013 B [22]	Implant site preparation	Prospective	18 MM 18 CTR	25 MM 25 CTR	6, 12, 24 m	Survival rate of 96% (both groups)	2 BPPV in CTR group
Crespi et al., 2014 A [23]	Ridge expansion	Prospective	23 MM 23 CTR	59 MM 59 CTR	6, 12, 24 m	Survival rate of 100% (MM) and 96.61% (CTR)	1 BPPV in CTR group
Crespi et al., 2014 B [24]	Dental extraction	Retrospective	156	427	-	No fracture or loss of cortical bone plate	-
Crespi et al., 2015 [25]	Ridge expansion	Retrospective	36	93	6, 12, 24 m	Survival rate of 98.92%	-
Crespi et al., 2016 A [26]	Implant site preparation	Prospective	40	40	36 m	Survival rate of 100%	-
Crespi et al., 2016 B [27]	Implant site preparation	Prospective	57	84	36 m	Survival rate of 100%	-
Crespi et al., 2017 [28] *	Dental extraction	Prospective	53	145	3 m	No signs of inflamed tissue or exposed bone	-
Menchini-Fabris GB, et al. 2020 A [29] *	Dental extraction	Retrospective	47	47	24–36 m	No postoperative complications	-
Menchini-Fabris GB, et al. 2020 B [30]	Sinus lift	Observational	29 MM 27 CTR	29 MM 27 CTR	36 m	Survival rate of 100% (MM) and 92.6% (CTR)	-
Crespi et al. 2021 [31]	Sinus lift	Retrospective	40	40	36 m	Survival rate 100%	-
Bruschi et al. 2021 [32]	Sinus lift	Retrospective	52	71	9–66 m (mean 29.8 m)	Survival rate 98.6%	-

Legend: MM, magnetic mallet; CTR, control group; m, months; BPPV, benign paroxysmal positional vertigo. * MM was not the main topic investigated in the manuscript.

## Data Availability

The data presented in this study are available on request from the corresponding author.

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
