# Peer review of "Usefulness of Magnetic Mallet in Oral Surgery and Implantology: A Systematic Review"

_jpm, 2022, doi:10.3390/jpm12010108_

Round 1

Reviewer 1 Report

Good paper.

However, one of the limitations is the limitation of the studies using MM in clinical patients.  Most of the papers cited were from a group of Crespi R, Capparè P, and Gherlone E. If more data from several groups of researchers are available, its significance will be greater.

Author Response

Author's Reply to the Review Report (Reviewer 1)

Thanks for your comment. We have tried to improve the quality of the manuscript.

Thanks in advance for your time to read this revised version of our manuscript.

Reviewer 2 Report

The authors aimed to summarize the current evidence regarding the Usefulness of magnetic mallet in oral surgery and implantology using a systematic review.

The manuscript is interesting and well written. However, there are critical points that need be addressed.

Major comments:

Authors mentioned that they used Jaded scale to assess the quality of studies, but they did not report a table summarizes the quality assessment. Quality assessment is critical and should be included and discussed. Authors should use the proper quality assessment tool for RCTs (Jaded or Cochrane) or observational studies (STROBE). Please check the link below: https://www.equator-network.org/reporting-guidelines/strobe/

“The significance of the relationship between the use of magnetic mallet and outcomes/complications of interventions was assessed with Fisher’s exact test (FT) to evaluate eventual differences between the test and control groups.” This is not clear and raise my concerns about the validity of the analysis performed. How you could combine the results of clinical prospective studies and observational studies? What type of control used in each study? Authors need to provide more details how they did the statistics and provide a copy the raw data as a supplementary file.

Authors should report type of controls in all the studies.

“Clinical trials comparing magnetic mallet with traditional intervention were evaluated with the Jadad scale [18] before being included in the meta-analysis; papers with a Jadad score of 3 or less were excluded from quantitative synthesis. What type of meta analysis was performed?.

Minor comments:

The abbreviation used in Table 1 are not common (e.g. S PS, D, etc.)  and could be replaced by text instead.

Author Response

Author's Reply to the Review Report (Reviewer 2)

Thank you for your consideration and suggestions.

We revised the paper following your recommendations to enhance article readability. Changes are highlighted in in the text using MS Word “Track Changes” function.

  • The authors aimed to summarize the current evidence regarding the Usefulness of magnetic mallet in oral surgery and implantology using a systematic review.

The manuscript is interesting and well written. However, there are critical points that need be addressed.

Thanks for your positive comment regarding the manuscript.

  • Major comments:

Authors mentioned that they used Jaded scale to assess the quality of studies, but they did not report a table summarizes the quality assessment. Quality assessment is critical and should be included and discussed. Authors should use the proper quality assessment tool for RCTs (Jaded or Cochrane) or observational studies (STROBE). Please check the link below: https://www.equator-network.org/reporting-guidelines/strobe/

Thanks for your comment. No randomized clinical trials were found after literature search, therefore quality assessment of studies included was not performed. The sentences were removed from the manuscript.

  • “The significance of the relationship between the use of magnetic mallet and outcomes/complications of interventions was assessed with Fisher’s exact test (FT) to evaluate eventual differences between the test and control groups.” This is not clear and raise my concerns about the validity of the analysis performed. How you could combine the results of clinical prospective studies and observational studies? What type of control used in each study? Authors need to provide more details how they did the statistics and provide a copy the raw data as a supplementary file.

Thanks for your comment. A review of the data allowed us to identify critical issues inherent in the different interventions typology compared during statistical analysis concerning the outcomes and complications. Therefore, by mutual agreement, the authors have decided to remove the references of the statistical analysis (Fisher's test) and focus only on the discussion of descriptive statistical analysis results.

  • Authors should report type of controls in all the studies.

Thanks for your comment. Not much information is available in the text of the articles reviewed: both in the test and control groups, patients required the same implant-prosthetic rehabilitation interventions.

  • “Clinical trials comparing magnetic mallet with traditional intervention were evaluated with the Jadad scale [18] before being included in the meta-analysis; papers with a Jadad score of 3 or less were excluded from quantitative synthesis. What type of meta analysis was performed?.

Thanks for your comment. No randomized clinical trials were found after literature search, therefore quality assessment of studies included was not performed. The sentences were removed from the manuscript.

  • Minor comments:

The abbreviation used in Table 1 are not common (e.g. S PS, D, etc.)  and could be replaced by text instead.

Thanks for your comment. The abbreviation used in Table 1 were removed replaced by text instead.

Thanks in advance for your time to read this revised version of our manuscript.

Round 2

Reviewer 2 Report

The authors addressed most of my comments and the manuscript has been improved considerably. However, authors did not add the quality assessment table of observational or non RCT studies based on strobe guidelines. Its critical in SR to include quality assessment table. Please check the link below: https://www.equator-network.org/reporting-guidelines/strobe/. 

Author Response

Thank you for your consideration and suggestions.

As reported in the reviewer's link ("https://www.equator-network.org/reporting-guidelines/strobe/"), STROBE is a checklist of items that should be included in reports of observational studies but there is not a score system to evaluate the quality of manuscript. Jadad or Newcastle Ottawa Scale are score systems to evaluate quality of manuscript to be included in metanalysis, but in this manuscript there are not articles to be compared in the quantitative analysis. Anyway, in the table 1 there are all the specific features and outcomes of the studies included in this systematic review.

Thanks in advance for your time to read this revised version of our manuscript.